

PeerJ Hubs
Published on behalf of

International Association for Biological Oceanography
IABO

# Marine aquaculture as a source of propagules of invasive fouling species

Daniel M. Lins[1] and Rosana M. Rocha[2]

[1] Ecology and Conservation Graduate Program, Universidade Federal do Paraná, Curitiba, Paraná, Brazil
[2] Zoology Department, Universidade Federal do Paraná, Curitiba, Paraná, Brazil

## ABSTRACT

Non-indigenous species tend to colonize aquaculture installations, especially when they are near international ports. In addition to the local environmental hazard that colonizing non-indigenous species pose, they can also take advantage of local transport opportunities to spread elsewhere. In this study, we examined the risk of the spread of eight invasive fouling species that are found in mussel farms in southern Brazil. We used ensemble niche models based on worldwide occurrences of these species, and environmental variables (ocean temperature and salinity) to predict suitable areas for each species with three algorithms (Maxent, Random Forest, and Support Vector Machine). As a proxy for propagule pressure, we used the tonnage transported by container ships from Santa Catarina (the main mariculture region) that travel to other Brazilian ports. We found that ports in the tropical states of Pernambuco, Ceará, and Bahia received the largest tonnage, although far from Santa Catarina and in a different ecoregion. The ascidians *Aplidium accarense* and *Didemnum perlucidum* are known from Bahia, with a high risk of invasion in the other states. The bryozoan *Watersipora subtorquata* also has a high risk of establishment in Pernambuco, while the ascidian *Botrylloides giganteus* has a medium risk in Bahia. Paraná, a state in the same ecoregion as Santa Catarina is likely to be invaded by all species. A second state in this region, Rio Grande do Sul, is vulnerable to *A. accarense*, the barnacle *Megabalanus coccopoma*, and the mussel *Mytilus galloprovincialis*. Climate change is changing species latitudinal distributions and most species will gain rather than lose area in near future (by 2050). As an ideal habitat for fouling organisms and invasive species, aquaculture farms can increase propagule pressure and thus the probability that species will expand their distributions, especially if they are close to ports. Therefore, an integrated approach of the risks of both aquaculture and nautical transport equipment present in a region is necessary to better inform decision-making procedures aiming at the expansion or establishment of new aquaculture farms. The risk maps provided will allow authorities and regional stakeholders to prioritize areas of concern for mitigating the present and future spread of fouling species.

Corresponding author
Rosana M. Rocha, rmrocha@ufpr.br

## INTRODUCTION

Marine aquaculture is growing faster than many other agricultural sectors, and can make up for fishery shortfalls and supply protein for many around the world (*Food and Agriculture Organization of the United Nations (FAO), 2018*). While the intentional introduction of non-indigenous marine species (NIMS) for cultivation should be preceded by a comprehensive risk analysis to minimize spillover and spread of NIMS (Code of Conduct on Responsible Fisheries, Article 9 on Aquaculture Development; *Food and Agriculture Organization of the United Nations (FAO), 1995*), indirect negative impacts caused by associated species are poorly understood (*Suplicy et al., 2015*) and not the object of evaluation in risk analysis.

Biofouling of aquaculture infrastructure (establishment of other species on the cultivated species and associated structures) increases management costs and decreases profitability (*Fitridge et al., 2012*), largely as a consequence of the cost of removal of unwanted species and yield reduction of the commercial species (*Bannister et al., 2019*). Biofouling varies spatially and temporally with community structure among regions (*Dürr & Watson, 2010*). When fouling species are also non-indigenous invasive species, aquaculture farms can act as stepping-stones allowing establishment and spread along adjacent coastal areas (*Ramsay et al., 2008*), and so they share responsibility for the negative environmental impacts caused by those invasive species in natural communities (*Simpson, Wernberg & Mcdonald, 2016*; *Robinson et al., 2007*).

Fouling non-indigenous species are primarily introduced by merchant ships that travel between oceans (*Hewitt & Campbell, 2010*) and by recreational yachts traveling regionally and along continuous coasts (*Peters, Sink & Robinson, 2017*; *Ulman et al., 2019*). The fundamental propagule pressure in a given region and the likelihood of invasion are determined by the rate at which a species is transported as biofouling (in hydrodynamically protected niche areas on ship hulls, ballast tanks, and sea chests; *Coutts & Dodgshun, 2007*; *Hulme, 2009*; *Davidson et al., 2010*). Transport by such means increases the probability of multiple invasions and reduces the effects of demographic and environmental stochasticity in recipient regions (*Simberloff, 2009*). To become invasive, a species must overcome a number of ecological and environmental barriers, even with many introduction events, and in general, only a fraction of the introduced species succeeds in becoming invasive (*Blackburn et al., 2011*). Continuously available new hard substrates for settlement (ropes, buoys, piers, and shells, all found in aquaculture facilities) enhance the probability of invasive species establishment (*McKindsey et al., 2007*). In addition, shellfish farms are often extensive in high-value, multiple-use, coastal areas (*e.g.*, important and significant habitats for conservation of biodiversity), where ports, marinas, tourism, recreation, and commercial fisheries are all found together, increasing propagule pressure and multiple introduction events (*Minchin, 2007*; *Castro, Fileman & Hall-Spencer, 2017*).

Mussel farming is very important to the local economy in southern Brazil (*Suplicy et al., 2015*). The cold and nutrient-rich South Oceanic Central Water is important and has favored aquaculture development in this region (*Lopes et al., 2006*). Currently, sheltered

bays in the state of Santa Catarina are responsible for over 95% of all mussel and oyster production in Brazil, with mussel and oyster farms established along most of the coast (*Santos & Della-Giustina, 2017*). The region is also notable for having the second largest concentration of shipyards and two of the nation's largest ports, to the north (Port of Itajaí) and to the south (Port of Imbituba) of the farming areas (*Agência Nacional de Transportes Aquaviários (ANTAQ), 2021*), which increases the risks of unintentional regional and international transport of fouling species. Non-indigenous invasive species are already prevalent fouling species in marine aquaculture facilities (*Rocha et al., 2009*; *Lins & Rocha, 2022*). Therefore, risk assessment of the impact of shellfish farming on the regional biodiversity should also consider fouling NIMS, their range expansion, and predictions of future movements of those species that are already known to be problematic in other regions.

Global regulations began to be enforced in 2017 to control marine bioinvasions and require all ships in international traffic to manage their ballast water and sediments (BWM Convention, *International Maritime Organization (IMO), 2004*), yet domestic port-to-port movement of species is still not the focus of biosecurity management in many countries, including Brazil, with long coastlines. Most of the 17 coastal states of Brazil have at least one major port that receives cargo ships from Santa Catarina. This suggests that the NIMS in aquaculture facilities in Santa Catarina have many opportunities for transport (strong propagule pressure) along the coast. In this study we focused on eight invasive NIMS established in aquaculture facilities at Santa Catarina (*Lins & Rocha, 2022*) to understand their potential to be introduced and invasive in other places in the country, based on current domestic shipping trade between ports in Santa Catarina and other states, using species suitability models to address current and future environmental suitability.

## MATERIALS AND METHODS

### Species and occurrences

Eight invasive NIMS recognized to be spread and abundant in aquaculture facilities in Santa Catarina (*Lins & Rocha, 2022*) were targeted in the present study: the ascidians *Aplidium accarense* (Millar, 1953), *Botrylloides giganteus* (Pérès, 1949), *Didemnum perlucidum* Monniot F., 1983 and *Styela plicata* (Lesueur, 1823), the bryozoans *Schizoporella errata* (Waters, 1878) and *Watersipora subtorquata* (d'Orbigny, 1852), the barnacle *Megabalanus coccopoma* (Darwin, 1854) and the bivalve *Mytilus galloprovincialis* Lamarck, 1819. These species are worldwide hitchhikers that have been unintentionally introduced to Brazil (although it is not clear if *A. accarense* is a regional or international invader).

Native and introduced occurrences of these target species were obtained from the global geographic distribution databases (OBIS.org and GBIF.org) and from published taxonomic and ecological studies obtained by searching for the valid names and resolved synonyms of the species in the Web of Science and Google Scholar portals. When studies reported general locations instead of precise geographic coordinates of the register, we acquired those coordinates using Google Maps, considering the existence of ports, marinas, and rocky shores as sites with the highest probability of presence in the location

mentioned in the study. Finally, we plotted species occurrences on maps and excluded registers without coordinates and outliers, that is, points in the continent.

## Current and future environment predictors

We modeled environmental suitability using variables from the Bio-Oracle v2.0 dataset (*Assis et al., 2018*), which has global marine layers with spatial resolution associated with a grid of cells of approximately 9 km$^2$ (5 arc-min) of average values for the period 2000–2014. We selected variables associated with seawater temperature and salinity which are considered the main drivers of the distribution of marine invertebrates (*Hauton, 2016*; *Whiteley & Mackenzie, 2016*) and which are also used to model the future scenarios of gas emissions. To control for strongly correlated environmental layers (r > 0.7), we systematically selected one and dropped the other in each pair of correlated variables (*Dormann et al., 2013*), resulting in a final set of four predictors that comprised maximum sea surface temperature, minimum sea surface salinity and ranges of both variables. Future (2050) projections of the same environmental predictors were obtained under two representative concentration pathways (RCP) for greenhouse gas emissions scenarios: one with a low greenhouse gas emission rate (RCP2.6) and one with a high greenhouse gas emission rate (RCP6.0) (see *Assis et al., 2018* for details).

## Ecological niche modeling

After gathering occurrence locations (presence) and environmental parameters in those locations we used the ENMTML package (*Andrade, Velazco & de Marco, 2020*) to fit the species current environmental suitability and projected models of future environmental conditions for the Brazilian coast. In the models, we employed the ensemble of the following algorithms: maximum entropy with default tuning (MXS or MaxEnt) (*Phillips, Anderson & Schapire, 2006*), random forest (RDF) (*Prasad, Iverson & Liaw, 2006*), and support vector machine (SVM) (*Guo, Kelly & Graham, 2005*). To reduce the effects of sampling bias, we randomly filtered species occurrences considering one presence only within each grid cell of a grid with a grain 2× the resolution of the environmental variables. It is a simple procedure with good performance (*Fourcade et al., 2014*). We used an absence ratio of one to 10 presences, which were randomly allocated within the lowest suitability areas predicted by a Bioclim model (*Engler, Guisan & Rechsteiner, 2004*), inside the area accessible to each species delimited by the Exclusive Economic Zone (*i.e.*, within 370 km of the coast). Models were validated by random bootstrap partition between 70% of the occurrence records for model training and 30% for testing the results (*Fielding & Bell, 1997*), with which we then evaluated the distributional models using True Skill Statistics (TSS > 0.8). We repeated this procedure 10 times for each algorithm and used the suitability value that maximizes the TSS to transform each map into a binary map with either presence or absence per cell (*Allouche, Tsoar & Kadmon, 2006*). Final models were constructed by an ensemble of all the algorithms using the average of suitability values weighted by the performance of the algorithms (TSS) (*Thuiller et al., 2009*). Environmentally suitable cells were categorized using a gradient from deep blue to red to

indicate low to high suitability for each species. All the procedures were performed in R 4.0.0 (*R Core Team, 2020*).

## Connectivity between ports

Merchant shipping is the main vector of transport and introductions of juvenile or adult marine organisms either by ballast water, hull fouling, or sea chests (*Coutts & Dodgshun, 2007*; *Hewitt, Gollasch & Minchin, 2009*). Container ships moved 97% of the cargo from Santa Catarina to other states in Brazil according to the data acquired online at the Brazilian national aquatic transport agency (ANTAQ). In the absence of data on the number of voyages and ships, we used the total tonnage of goods transported during 5 years (2015 to 2019) as a surrogate of the connectivity between states. We ranked states comparatively in three categories: high (>1,000 thousand tons), intermediate (100–1,000 thousand tons), and low connectivity (<100 thousand tons) with Santa Catarina.

## Risk assessment

To assess the risk of species transport and introduction/invasion (connectivity + suitability), we built a matrix that overlaps the information on cargo transport from Santa Catarina to each Brazilian coastal state, and environmental suitability for each of the eight focus species (Table S1). The joint assessment of environmental suitability in potential recipient regions, complemented by information on vectors of transport, has previously been used to forecast species introduction risk (*Goldsmit et al., 2018*; *Lins et al., 2018*). In our assessment, we considered connectivity (propagule pressure) as more important than environmental adequacy because there have been numerous situations where species can rapidly adapt to new environmental situations and expand their known niche (*Broennimann et al., 2007*; *Early & Sax, 2014*). Following this reasoning, states with high connection were not classified with a low risk even where environment adequacy was very low, and accordingly, states with low connection were not classified with a high risk, even where environment adequacy was high (Table S1).

## RESULTS

The bryozoans *Schizoporella errata* and *Watersipora subtorquata* were found to be the most widespread species, already present in 10 and eight states, respectively, between Santa Catarina (SC) and Ceará (CE). The ascidians *Aplidium accarense*, *Didemnum perlucidum*, *Styela plicata*, and the barnacle *Megabalanus coccopoma* are in six states, from Santa Catarina to Bahia, with the exception of *A. accarense* (found through Rio Grande do Norte—RN). The ascidian *B. giganteus* occurs in four states closer to Santa Catarina, and the mussel *M. galloprovincialis* occurs in Santa Catarina only (Fig. 1).

Santa Catarina delivered a total of 10.758 thousand tons in goods from 2015 to 2019, with a remarkable increase in containerized cargo (from 1,113 to 2,351 thousand tons in 5 years), and now accounts for 97% of all goods shipped from Santa Catarina to elsewhere in Brazil. The northeastern states of Pernambuco, Ceará, and Bahia were the main destinations and more prone to receive propagules from Santa Catarina (Fig. 1). The states with intermediate connection to Santa Catarina were São Paulo, Espírito Santo, Rio de

**Figure 1** Tonnage (thousand tons) transported by container ships from Santa Catarina to other states in Brazil over 5 years (2015–2019), current occurrences (*), and environmental suitability (colored cells).

| Federation states | Total (x1000) | Aplidium accarense | Botrylloides giganteus | Didemnum perlucidum | Styela plicata | Schizoporella errata | Watersipora subtorquata | Megabalanus coccopoma | Mytilus galloprovincialis |
|---|---|---|---|---|---|---|---|---|---|
| Pernambuco (PE) | 3083 | yellow | | red | | yellow * | red | | |
| Ceará (CE) | 2220 | blue | | red | | blue * | yellow * | | |
| Bahia (BA) | 1336 | red * | blue | red * | blue * | red * | red * | blue * | |
| São Paulo (SP) | 752 | red * | red * | red * | red * | red * | red * | red * | blue |
| Rio Grande do Sul (RS) | 597 | red | yellow | blue | yellow | yellow | blue | red | red |
| Rio de Janeiro (RJ) | 360 | red * | red * | red * | red * | red * | red * | yellow * | yellow |
| Espírito Santo (ES) | 327 | red * | yellow * | red * | yellow * | red * | yellow * | yellow * | blue |
| Paraná (PR) | 236 | red | red | red * | red * | red | red | red * | red |
| Paraíba (PB) | 13 | yellow | | red | | yellow * | red | | |
| Rio Grande do Norte (RN) | 4 | blue * | | red | | yellow * | yellow * | | |
| Maranhão (MA) | 3 | blue | | yellow | | | | | |
| Sergipe (SE) | 0 | yellow | | yellow | | blue | red | | |
| Alagoas (AL) | 0 | yellow | | red | | yellow * | yellow * | | |
| Piauí (PI) | 0 | blue | | red | | | blue | | |
| Amapá (AM) | 0 | | | | | | | | |

Red cells, the greatest environmental suitability; yellow, medium suitability; blue, low suitability; uncolored, not suitable.

Janeiro, in the southeast, and Paraná and Rio Grande do Sul in the south. The least-connected group of states was Paraíba, Rio Grande do Norte and Maranhão in the northeast. Amapá, Alagoas, Sergipe, and Piauí did not receive container ships from Santa Catarina during the time interval under study. Pará received container ships but the main port is in freshwater, thus it was not considered in the analysis.

The number of unique occurrences per species used for modeling ranged from 20 to 678, with accurate predictions (TSS > 0.8) and little variation (Table S2). The evaluation index indicated that the SVM and RDF models performed similarly and were more accurate than MXL (Table S2). Except for three species under the RDF model, temperature variables were consistently the main drivers of predictive performances across algorithms (Table 1). The ensemble of the models showed that there are suitable areas not yet occupied to which species can expand their distribution, both currently and under future global warming scenarios (Figs. 1–4). Environmental conditions seem to be similar between both RCP 2.6 and 6.0 scenarios, and predicted species distribution is almost the

**Table 1 Summary of contributions (as %) of environmental variables used for predicting performance of species suitability models.**

| Species | MXS[1] | | | | RDF | | | | SVM | | | |
|---|---|---|---|---|---|---|---|---|---|---|---|---|
| | SSS | | SST | | SSS | | SST | | SSS | | SST | |
| | Min | Range | Max | Range | Min | Range | Max | Range | Min | Range | Max | Range |
| *Aplidium accarense* | 25 | 2 | 29 | **42** | **32** | 18 | 26 | 22 | 24 | 4 | 22 | **50** |
| *Botrylloides giganteus* | 25 | 11 | **33** | 31 | 26 | 26 | 17 | **28** | 19 | 7 | 26 | **47** |
| *Didemnum perlucidum* | 23 | 5 | **67** | 5 | 22 | 18 | **31** | 29 | 28 | 3 | **64** | 5 |
| *Styela plicata* | 25 | 8 | 27 | **50** | 28 | 14 | **37** | 21 | 22 | 11 | 31 | **36** |
| *Schizoporella errata* | 25 | 3 | 27 | **43** | **38** | 16 | 21 | 23 | 23 | 2 | 35 | **38** |
| *Watersipora subtorquata* | 28 | 11 | **42** | 18 | **38** | 7 | 29 | 24 | 26 | 11 | **39** | 22 |
| *Megabalanus coccopoma* | 8 | 7 | 42 | **43** | 25 | 7 | 31 | **37** | 15 | 5 | **55** | 25 |
| *Mytilus galloprovincialis* | 20 | 13 | 5 | **62** | 19 | 10 | **51** | 21 | 16 | 12 | 3 | **69** |

Notes:
[1] MXS, MaxEnt; RDF, random forest and SVM, support vector machine.
Sea surface salinity (SSS) minimum and range, sea surface temperature (SST) maximum and range obtained from Bio-Oracle v2.0 database. Maximum values for each species and model in bold.

same, thus RCP 2.6 only is shown here and RCP 6.0 can be found in Supplemental Material (Figs. S1–S3).

Among the highly-connected states, Pernambuco is currently environmentally suitable for *A. accarense*, *D. perlucidum*, and *W. subtorquata*, while Ceará is suitable for *D. perlucidum* and marginally for *A. accarense*, and Bahia has low suitability for *B. giganteus* (Fig. 1). Among the intermediately-connected states, São Paulo, Rio de Janeiro, and Espírito Santo have already been colonized by all species with the exception of *M. galloprovincialis*. Rio de Janeiro has intermediate environmental suitability for this species while the other two states have low suitability. Paraná has high suitability for all species that have not yet colonized it, and Rio Grande do Sul has variable suitability for those eight species. Both low-connected states are very suitable for *D. perlucidum* and *W. subtorquata*, and Paraíba is moderately suitable for *A. accarense*.

Considering the connectivity and environmental suitability together (Figs. 2–4), in the northeast, Pernambuco (PE) is the state most at risk, and is likely to receive propagules and be invaded by *A. accarense*, *D. perlucidum* and *W. subtorquata* populations coming from Santa Catarina (SC). Ceará (CE) is at high risk of invasion by *D. perlucidum* and moderate risk by *A. accarense*. The other states are at a medium risk of invasion by *B. giganteus* (Bahia—BA), *D. perlucidum* (Paraíba—PB, Rio Grande do Norte—RN) and *W. subtroquata* (Paraíba), and low risk by *A. accarense* (Paraíba). In the southeast, only Rio de Janeiro (RJ) is at moderate risk of being invaded by *M. galloprovincialis*, while São Paulo (SP) and Espírito Santo (ES) both are at low risk. In the south, Paraná (PR) is at high risk of invasion by *A. accarense*, *B. giganteus, S. errata, W. subtorquata* and *M. galloprovincialis*. Rio Grande do Sul (RS) is at high risk of invasion by *A. accarense*, *M. coccopoma*, and *M. galloprovincialis*, at a moderate risk by *B. giganteus, S. errata*, and *W. subtorquata*, and at low risk by *D. perlucidum* and *S. plicata*.

In the future, all species (with the exception of *A. accarense and D. perlucidum*) are likely to extend their distributions towards lower latitudes and invade more in the

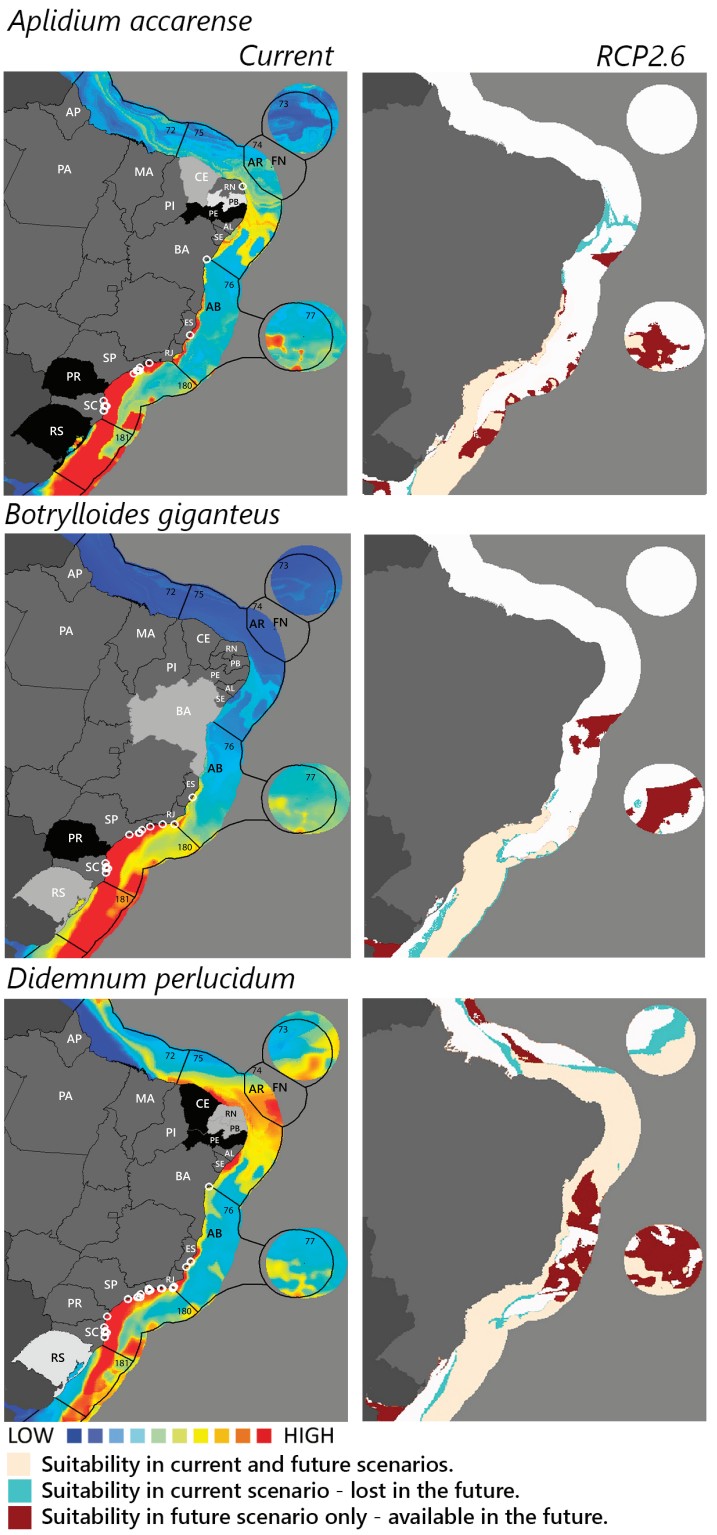

**Figure 2 Maps of coastal Brazil and main oceanic island regions in which current and future species distributions are projected for *Aplidium accarense, Botrylloides giganteus,* and *Didemnum perlucidum.*** Current = species occurrences (white circles), *Spalding et al. (2007)* ecoregions (solid lines), and the states (abbreviations as in Fig. 1) at most risk of introduction and invasion, based on both connectivity and environment suitability (black, high risk; grey, medium risk; light grey, low risk). RCP

**Figure 2 (continued)**
2.6 = Future climate scenario in the year 2050. Environmental suitability maps were generated by the ensemble procedure of three types of Ecological Niche Models (MaxEnt, Support Vector Machine, and Random Forest). AB, Abrolhos bank; AR, Atol das Rocas; FN, Fernando de Noronha. Ecoregions: 72, Amazonia; 73, Sao Pedro and Sao Paulo Islands; 74, Fernando de Naronha and Atoll, das Rocas; 75, Northeastern Brazil; 76, Eastern Brazil; 77, Trindade and Martin Vaz Islands; 180, Southeastern Brazil; 181, Rio Grande.                                     

northeast, while *B. giganteus* will expand only in the RPC 6.0 scenario, and *M. galloprovincialis* only in the RPC 2.6 scenario (Figs. 2–4 and Figs. S1–S3).

## DISCUSSION

Most invasive species in this study are already found along much of the Brazilian coast, from Santa Catarina to Bahia. *Mytilus galloprovincialis* is an exception and a relatively recent invasion, so it is still restricted to the shellfish farms of Santa Catarina. Species distribution models indicated that environmentally suitable regions exist but have not yet been invaded, both in the south and northeastern regions (Fig. 1). We do not yet understand whether the arrival of these species is simply a matter of time, whether propagule transport is too low or nonexistent, whether there are any other biological or environmental constraints on their establishment, or finally, whether they are absent as an artifact of the lack of good monitoring programs. A combination of causes is likely. For example, the lack of propagules from Santa Catarina, suitable hard substrates for attachment, and good marine biodiversity monitoring programs could be the combined causes to explain why Sergipe and Piauí currently do not have any of the focus species in this study.

Landscape features were not accounted for in the environmental suitability models we used, but they may be important to understand community assembly and species presence. One important driver for sessile NIMS is the availability of hard substrates for attachment (*Ruiz et al., 2009*). For example, Rio Grande do Sul (RS) has a coastal landscape characterized by long sandy beaches and Quaternary beachrock reefs running at ~40 m deep along the north coast of the state (*Bergue et al., 2022*). Besides this formation, the only hard surfaces available is this region are artificial walls in the entrance of the Patos Lagoon where the port is located. Thus, this is another example that combines a lack of hard substrates for attachment and a lack of biota surveys, which could explain the absence of the focal species, despite the environmental suitability for most species, closeness, and intermediate connectivity with Santa Catarina.

Predation may also limit species distributions and is not included in climate suitability models even though predation is known to have a strong influence on the composition of benthic communities in different latitudes and increasingly towards the tropics (*Dias et al., 2020*, *Freestone et al., 2021*). For instance, predation on ascidians usually liberate space for organisms protected by calcareous exoskeleton such as bryozoans and barnacles, favoring invasive species in those groups (*Kremer & Rocha, 2016*, *Dias et al., 2020*).

Temperature was the most important driver for suitable areas and it is known to interact with or influence other drivers. Primary productivity, for instance, is positively correlated

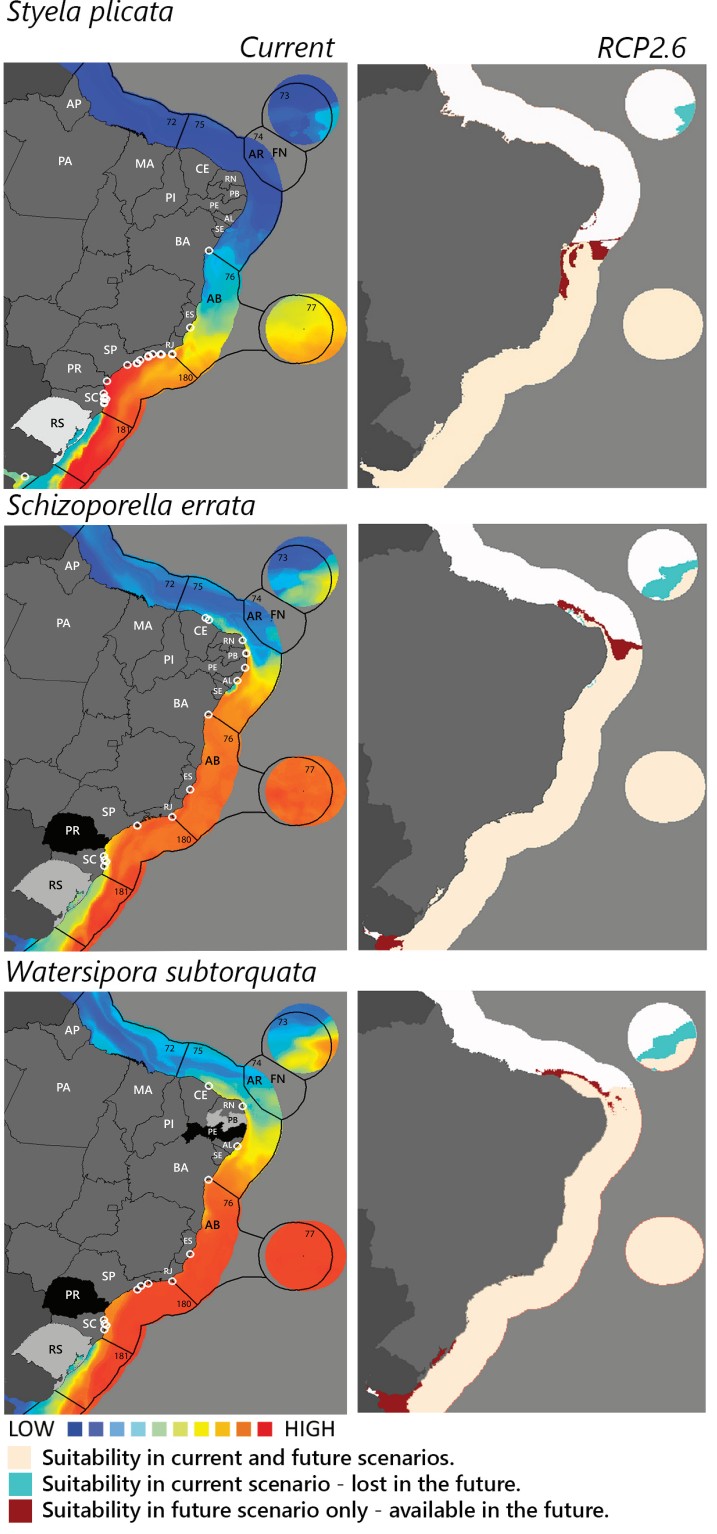

**Figure 3 Maps of coastal Brazil and main oceanic island regions in which current and future species distributions are projected for *Styela plicata*, *Schizoporella errata*, and *Watersipora subtorquata*.** Current = species occurrences (white circles), *Spalding et al. (2007)* ecoregions (solid lines), and the states (abbreviations as in Fig. 1) at most risk of introduction and invasion, based on both connectivity and environment suitability (black, high risk; grey, medium risk; light grey, low risk). RCP 2.6 = Future
**Figure 3** (continued)
climate scenario in the year 2050. Environmental suitability maps were generated by the ensemble procedure of three types of Ecological Niche Models (MaxEnt, Support Vector Machine, and Random Forest). AB, Abrolhos bank; AR, Atol das Rocas; FN, Fernando de Noronha. Ecoregions: 72, Amazonia; 73, Sao Pedro and Sao Paulo Islands; 74, Fernando de Naronha and Atoll, das Rocas; 75, Northeastern Brazil; 76, Eastern Brazil; 77, Trindade and Martin Vaz Islands; 180, Southeastern Brazil; 181, Rio Grande.                                             

with seawater temperature and eutrophication has already been associated with bioinvasion by ascidians (*Marins et al., 2010*). Context-dependent effects can also be expected. In polluted waters, competition between the bryozoans *S. errata* and *W. subtorquata* is strengthened at higher temperatures (*McKenzie, Brooks & Johnston, 2011*). Santa Catarina is at a higher latitude, in a different marine realm and province (*sensu Spalding et al., 2007*) than the tropical states in the country, yet four species (*A. accarense, D. perlucidum, S.errata*, and *W. subtorquata*) have such a current broad distribution that niche models showed that those tropical states were environmentally adequate for them. Global warming is expected to generate range shifts of NIMS towards higher latitudes (*Sorte, Williams & Carlton, 2010*; *Canning-Clode & Carlton, 2017*), however, our predictions indicate that in global warming scenarios, most species will gain suitable areas towards the equator. This suggests that for invasive species range shift predictions could be more complex, given that widespread invasive species have high phenotype plasticity, allowing them to occupy climatic niches distinct from those they occupy in the regions of origin (*Broennimann et al., 2007*; *Rocha, Castellano & Freire, 2017*). We used registers of occurrences of native and introduced ranges without distinction to calibrate and test the models, which usually perform better than when using occurrences from the native range only because they better reflect phenotype variation of the species of interest (*Broennimann & Guisan, 2008*).

Propagule pressure and environmental suitability make *A. accarense, D. perlucidum*, and *W. subtorquata* the species with the greatest risk of invasion of tropical regions. The first is already widespread in different continents (*Monniot, 1969*; *Rocha et al., 2010*; *López-Legentil et al., 2015*) but as of yet, it has no known impact. *Didemnum perlucidum* is known to spread from artificial substrates to seagrass beds (*Halophila ovalis*) in Western Australia (*Simpson, Wernberg & Mcdonald, 2016*) with a possible impact on plant photosynthesis, and the abundance of seagrass-associated mud snails. In Santa Catarina mussel farms, this species reduces mussel yield (*Lins & Rocha, 2020*). *Watersipora subtorquata* is a bioengineer that can have variable impacts on the sessile and mobile species on hard substrate communities (*Scott & terHorst, 2020*).

The mussel *Mytilus galloprovincialis* merits concern and ranks among the 100 most invasive species worldwide (*Lowe et al., 2000*) with the important environmental impact of changing biodiversity in natural communities. This mussel was introduced for cultivation in South Africa from where it has spread to adjacent natural environments (*McQuaid & Phillips, 2000*) with impacts on habitat complexity and change in species dominance within the benthic community (*Sadchatheeswaran, Branch & Robinson, 2015*). Rio Grande do Sul and Paraná are environmentally suitable for the species, and, as neighbor states of Santa

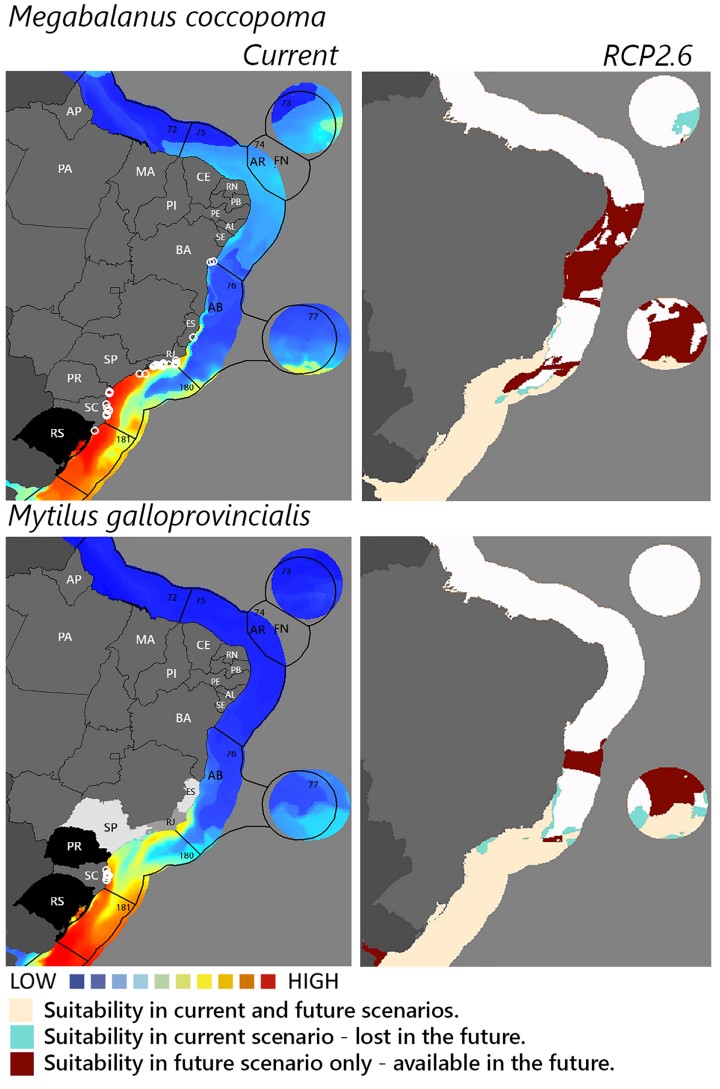

**Figure 4 Maps of coastal Brazil and main oceanic island regions in which current and future species distributions are projected for *Megabalanus coccopoma* and *Mytilus galloprovincialis*.** Current = species occurrences (white circles), *Spalding et al. (2007)* ecoregions (solid lines), and the states (abbreviations as in Fig. 1) at most risk of introduction and invasion, based on both connectivity and environment suitability (black, high risk; grey, medium risk; light grey, low risk). RCP 2.6 = Future climate scenario in the year 2050. Environmental suitability maps were generated by the ensemble procedure of three types of Ecological Niche Models (MaxEnt, Support Vector Machine, and Random Forest). AB, Abrolhos bank; AR, Atol das Rocas; FN, Fernando de Noronha. Ecoregions: 72, Amazonia; 73, Sao Pedro and Sao Paulo Islands; 74, Fernando de Naronha and Atoll, das Rocas; 75, Northeastern Brazil; 76, Eastern Brazil; 77, Trindade and Martin Vaz Islands; 180, Southeastern Brazil; 181, Rio Grande.

Catarina, are at high risk of invasion. Rio de Janeiro is the next state at risk, followed by São Paulo and Espírito Santo. Given the observed rapid adaptive genetic variation associated with temperature enabling this species to invade a wide range of thermal habitats successfully (*Han & Dong, 2020*) it is not surprising that our model of future warming scenario (RCP 2.6) predicts the establishment of *M. galloprovincialis* up to Bahia.

Managers should be aware that a continuous propagule supply from Santa Catarina could further intensify invasions by introducing adaptive genetic variation, even in states with these NIMS (*Ghabooli et al., 2013*). The main source of presence records of these NIMS is from Rapid Assessment Surveys or experimental studies carried out in marinas and ports, and so do not indicate that they are already established on natural substrates elsewhere. Increasing the abundance and genetic diversity of propagules could enhance the probability of invasion of natural communities. Managers should also consider that not only the shallow coast has an adequate environment for the NIMS here studied. Several human offshore activities, including domestic shipping routes, decommissioned ship sinking, industrial fishing, wind, oil and natural gas exploitation equipment, and submarine cables are associated with hard surfaces susceptible to fouling by NIMS acting as stepping stones for their dispersal and subsequent expansion to natural communities (*De Mesel et al., 2015*; *Gardner et al., 2016*).

Risk maps, such as those in this study, are valuable tools for decision-making to determine where NIMS introductions are most likely and where to allocate resources. Predictive models can help to determine the most probable pathways of population movement, help to choose sampling sites, and to reduce the costs of molecular studies to detect the origins of introduced species (*Pritchard, Stephens & Donnelly, 2000*; *Falush, Stephens & Pritchard, 2007*). Predictive models also have the advantage of using public information and have already successfully predicted marine species introduction elsewhere (*e.g.*, *C. lepadiformis* to Australia and *S. clava* to Argentina, *Lins et al., 2018*).

We cannot rule out the possibility that the ports themselves are the main source of propagules generating the risk of NIMS spread in this study. A comparative study to understand the relative importance of port and aquaculture farm populations as sources of propagules would be necessary. But the presence of established reproductive populations in aquaculture farms (*Lins & Rocha, 2022*) near ports calls for an integrated approach to informing decision-making procedures for expanding or establishing aquaculture farms. Stakeholder perceptions may vary about the importance of an ecosystem approach when locating aquaculture parks (*Vianna & Filho, 2018*), so these results should be made clear to them. Ideally, the location of aquaculture farms should avoid multiple-use areas (where recreational, fishing, and international shipping vessels increase the chance of dispersal). Few studies estimate the natural dispersal of NIMS from aquaculture farms, and that is required to determine the extent of buffer zones around them. Coastal shipping should also become the focus of biosecurity management and risk analysis, both national and international, in addition to concern with the international transport of NIMS.

## ACKNOWLEDGEMENTS

We would like to express our deepest gratitude to all the mussel farmers and Prof. Gilberto Manzoni with Univali Oceanography Department who have supported us in the field. We also thank Dr. Leandro M. Vieira who confirmed the identification of the bryozoan species, Roman Wenne who confirmed the identification of the mussel species, and James J. Roper who reviewed this text in its entirety.

### Funding

This research was funded by a research grant to RMR provided by the National Council for Scientific and Technological Development—Brazil (CNPq, 309295/2018-1) and a scholarship to DML provided by Coordenação de Aperfeiçoamento de Pessoal de Nível Superior—Brazil (CAPES, finance Code 001). The funders had no role in study design, data collection and analysis, decision to publish, or preparation of the manuscript.

### Grant Disclosures

The following grant information was disclosed by the authors:
National Council for Scientific and Technological Development—Brazil: 309295/2018-1.
Coordenação de Aperfeiçoamento de Pessoal de Nível Superior—Brazil: 001.

### Competing Interests

The authors declare that they have no competing interests.

### Author Contributions

- Daniel M. Lins conceived and designed the experiments, performed the experiments, analyzed the data, prepared figures and/or tables, authored or reviewed drafts of the article, and approved the final draft.
- Rosana M. Rocha conceived and designed the experiments, prepared figures and/or tables, authored or reviewed drafts of the article, and approved the final draft.

### Data Availability

The dataset of occurrences used to model species distribution and the script used in ENMTML package to create the models are available in the Supplemental Files.

### Supplemental Information

Supplemental information for this article can be found online at http://dx.doi.org/10.7717/peerj.15456#supplemental-information.

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
