# Peer review of "Marine aquaculture as a source of propagules of invasive fouling species"

_PeerJ, doi:10.7717/peerj.15456_

## Round 0.1 · original submission · Major Revisions

Dear authors

We received two detailed reviews of your work. Please address those comments in your review which should improve the quality of your work.

In addition, I have a few queries:

Methods

- Your maps are interesting but they do not indicate ecoregions, Brazilian coastal states and islands that are often mentioned in the text. This needs to be included if we want everyone to understand your work. It is also difficult to differentiate where the oceanic islands are in the maps (in some they are not indicated).

Lines 113-111 - please add to your supplemental table with the geographical coordinates, the source of data or cite the reviewed literature where the records have been obtained. Is there info on the substrate where species were recorded in the dataset (aquaculture ponds, natural substrates or other human structure) that could be added?

Discussion - I follow one of the reviewer's comments about the possibility of occurrence of invasive species in non-aquaculture substrates in the studied regions (indicated in the first sentence of your discussion section). The species may also be associated to these structures in one region but not in others given substrate availability. Could you please address this more clearly in your discussion?

Lines 227-231 - this is an important point, pseudo-absences of introduced species. Is there an implication of that to your models?

Lines 237-240 - what type of hard substrate is absent. Please cite references to support these sentences.

Lines 251-254 - Please clarify this sentence. Which correlated factors? What is the final effect on species invasion and how it changes across gradients? What "context-dependent shifts"mean?

Line 256-259 - Pleas clarify this sentence.

Line 290-294 - For management purposes it is important to understand where these species already occur and which of those can migrate northwards, and the likelihood of that happening. Please make this more clear in your discussion.

Line 295 - Which offshore habitats? Oceanic islands? please clarify and indicate which species may colonize each of those with its consequences.

Reviewer 1 ·

Basic reporting

This article has been well structured and carefully written. I have provided a number of edits in the pdf version to help with the English. The literature has largely been well covered. Nonetheless, the inclusion of the following papers would help to broaden the coverage by including very relevant recent works:

Mahanes & Sorte 2019 https://doi.org/10.21425/F5FBG40527 - Helpful regarding propagule pressure and the interaction with climate change in marine systems.

Robinson et al 2020 https://doi.org/10.3897/neobiota.62.55729 - Provides insights into indirect ways that climate change could impact the spread of species.

Mellin et al 2016 https://doi.org/10.1016/j.biocon.2016.11.008 - Will add to the discussion of factors not accounted for in the current study.

Experimental design

The aim was clear and the methods appropriately addressed this. The methods section was detailed and clear to enable repetition of the work that was done.

Validity of the findings

The required data were all provided and are accessible in the documents given. based on the sound methods the results provided are valid. My only concern is around the key conclusion that this study shows that aquaculture needs to be separated from ports. That statement has a logical basis but this study cannot be used to support or disprove it, because the design does not separate out the role of ports themselves vs the role of farms. To make this statement, the authors would need to compare the invasion risk of a busy port with near by farms with the invasion risk of a busy port without farms. Only that way would they be able to make the clear statement that farms are the source of the risk.

Additional comments

In places the paper is a little difficult for an outsider to follow because of the focus the local Brazilian context (e.g. reference to local geography). I think this can be addressed by framing this work as a Brazilian case study that addresses issues appliable elsewhere in the world. This need only be a subtle change, but will make the research more accessible to broader audience and get this good work out into the literature.

Please note that I have added a few minor comments into the pdf version of the manuscript.

Annotated reviews are not available for download in order to protect the identity of reviewers who chose to remain anonymous.

·

Basic reporting

The manuscript is well written and data very important to the theme of Bioinvasion and marine biodiversity and conservation on South Atlantic. Data about ports connection and environmental variables as sea surface temperature are of high importance on this field. Figures and tables are in high quality .

Experimental design

Experimental design are appropriate to hypothesis and related to data acquire.

Validity of the findings

No comment

Additional comments

My overall opinion is that manuscript is in high quality and only few changes are needed to highlight both the importance of Risk Assessment of species introduction on South Atlantic, mainly due to cabotage navigation that are not under the same Laws than international navigation. The second is to highlight the importance of the Barzilian Oceanic Islands, mainly the coral reef area of Abrolhos.

---

## Round 0.2 · Minor Revisions

Dear authors

Thank you for addressing most previous queries raised from the reviewers. However, before accepting this work for publication, please address the following:

1- an improved version of your maps is needed, as they are central in the interpretation of your results. I will stress again the need for a clear indication of all localities in the maps so non-brazilians can easily know the location of Brazilian states and marine ecoregions without having to refer to Spalding's work. Although they were drawed in some (but not all) the maps, it is still hard for a general audience to interpret them. Drawing of the ecoregions can be clearly improved. The port locations in the maps are also very difficult to set apart from each other, specially in SE Brazil, so please consider using alternative strategies as long as they make it easier to read. In addition, one can not easily see any differences between the RCP2.6 and 6.0 scenarios - they both look very similar, and some differences only appear in isolated patches of water offshore. So please consider changing the way those maps are presented so it is easier to compare both.

In addition, please clarify:

Discussion
Ln 241-242 - please cite which "environmentally suitable regions exist that have not been invaded"

Ln 250-260 - This whole paragraph is full of example locations that can only be understood by Brazilians. This need to be changed, along with the maps.

Ln 261-270 - Again, is predation important to control benthic communities only in Brazil or in Santa Catarina? Or is that an ecological process common to coastal ecosystems in general? Could you please reframe the sentences to make it more general to coastal marine ecosystems and less local?

Ln 280-282 - I can't see the range expansion towards higher latitudes in your work. Please explain in detail or use figures that support this sentence.


Congratulations on your work and I am looking forward in receiving your revised version.

---

## Round 0.3 · accepted · Accept

Dear authors

Thanks for addressing the comments raised.